# Novel Variants in *MPV17, PRX, GJB1*, and *SACS* Cause Charcot–Marie–Tooth and Spastic Ataxia of Charlevoix–Saguenay Type Diseases

**DOI:** 10.3390/genes14020328

**Published:** 2023-01-27

**Authors:** Qaiser Zaman, Muhammad Abbas Khan, Kalsoom Sahar, Gauhar Rehman, Hamza Khan, Mehwish Rehman, Ilyas Ahmad, Muhmmad Tariq, Osama Yousef Muthaffar, Angham Abdulrhman Abdulkareem, Fehmida Bibi, Muhammad Imran Naseer, Muhammad Shah Faisal, Naveed Wasif, Musharraf Jelani

**Affiliations:** 1Department of Zoology, Government Postgraduate College Dargai, Malakand 23060, Pakistan; 2Higher Education Department, Government of Khyber Pakhtunkhwa, Peshawar 24550, Pakistan; 3Department of Zoology, Abdul Wali Khan University, Mardan 23200, Pakistan; 4National Center for Bioinformatics, Quid-I-Azam University, Islamabad 45320, Pakistan; 5Institute for Cardiogenetics, University of Lübeck, DZHK (German Research Centre for Cardiovascular Research), Partner Site Hamburg/Lübeck/Kiel, and University Heart Centre Lübeck, 23562 Lübeck, Germany; 6Rare Diseases Genetics and Genomics, Centre for Omic Sciences, Islamia College, Peshawar 25120, Pakistan; 7Department of Pediatrics, Faculty of Medicine, King Abdulaziz University, Jeddah 21589, Saudi Arabia; 8Department of Biochemistry, Faculty of Science, King Abdulaziz University, Jeddah 21589, Saudi Arabia; 9Center of Excellence in Genomic Medicine Research, King Abdulaziz University, Jeddah 21589, Saudi Arabia; 10Special Infectious Agents Unit, King Fahd Medical Research Centre, King Abdulaziz University, Jeddah 21589, Saudi Arabia; 11Department of Medical Laboratory Technology, Faculty of Applied Medical Sciences, King Abdulaziz University, Jeddah 21589, Saudi Arabia; 12Institute of Human Genetics, Ulm University Medical Center, Ulm University, 89081 Ulm, Germany; 13Institute of Human Genetics, University Hospital Schleswig-Holstein, Campus Lübeck, 23538 Lübeck, Germany

**Keywords:** *MPV17*, *PRX*, *GJB1*, *SACS*, CMT, ARSACS, WES

## Abstract

Charcot–Marie–Tooth disease (CMT) and autosomal recessive spastic ataxia of Charlevoix–Saguenay type (ARSACS) are large heterogeneous groups of sensory, neurological genetic disorders characterized by sensory neuropathies, muscular atrophies, abnormal sensory conduction velocities, and ataxia. CMT2EE (OMIM: 618400) is caused by mutations in *MPV17* (OMIM: 137960), CMT4F (OMIM: 614895) is caused by *PRX* (OMIM: 605725), CMTX1 (OMIM: 302800) is caused by mutations in *GJB1* (OMIM: 304040), and ARSACS (OMIM: 270550) is caused by mutations in *SACS* (OMIM: 604490). In this study, we enrolled four families: DG-01, BD-06, MR-01, and ICP-RD11, with 16 affected individuals, for clinical and molecular diagnoses. One patient from each family was analyzed for whole exome sequencing and Sanger sequencing was done for the rest of the family members. Affected individuals of families BD-06 and MR-01 show complete CMT phenotypes and family ICP-RD11 shows ARSACS type. Family DG-01 shows complete phenotypes for both CMT and ARSACS types. The affected individuals have walking difficulties, ataxia, distal limb weakness, axonal sensorimotor neuropathies, delayed motor development, pes cavus, and speech articulations with minor variations. The WES analysis in an indexed patient of family DG-01 identified two novel variants: c.83G>T (p.Gly28Val) in *MPV17* and c.4934G>C (p.Arg1645Pro) in *SACS*. In family ICP-RD11, a recurrent mutation that causes ARSACS, c.262C>T (p.Arg88Ter) in *SACS*, was identified. Another novel variant, c.231C>A (p.Arg77Ter) in *PRX*, which causes CMT4F, was identified in family BD-06. In family MR-01, a hemizygous missense variant c.61G>C (p.Gly21Arg) in *GJB1* was identified in the indexed patient. To the best of our knowledge, there are very few reports on *MPV17*, *SACS*, *PRX*, and *GJB1* causing CMT and ARSACS phenotypes in the Pakistani population. Our study cohort suggests that whole exome sequencing can be a useful tool in diagnosing complex multigenic and phenotypically overlapping genetic disorders such as Charcot–Marie–Tooth disease (CMT) and spastic ataxia of Charlevoix–Saguenay type.

## 1. Introduction

Charcot–Marie–Tooth disease (CMT), or hereditary motor and sensory neuropathy (HMSN), was first reported in 1886 by three neurologists, Jean-Martin Charcot, Pierre Marie, and Howard Henry Tooth, and was referred to as peroneal muscular atrophy (PMA) [1,2]. With an estimated prevalence of 1 in 2500 worldwide, it is one of the most common forms of a heterogeneous group of genetic neuropathies that are characterized by abnormalities in the peripheral and central nervous system [3,4,5]. It typically appears in the first two decades of life, and the majority of patients with CMT have an autosomal dominant (AD) inheritance, but many have autosomal recessive (AR) or X-linked inheritance [6]. The disease results in a slowly progressive and length-dependent degeneration of the peripheral nerves, causing muscle weakness, numbness, reduced tendon reflexes, atrophy in the feet and legs (which later extends to the hands), and slight to moderate distal sensory impairment [7]. Patients may also exhibit hip dysplasia, learning loss, and foot deformities such as pes cavus (high-arched foot deformity) [3,8,9]. In addition to these and other symptoms (Table 1) that vary depending on the different CMT subtypes, the disease can greatly decrease the quality of life of those affected [10,11]. 

CMT is classified into six main types: CMT1, CMT2, CMT3, CMT4, CMTX, and I-CMT (DI-CMT and RI-CMT), which are further subdivided into subtypes (Appendix A). Molecular analysis has identified over 50 genes involved in these 76 subvarieties of the disease, which control specific functions in sub-neurons or Schwann cells, such as signaling, transduction, and nerve impulse propagation. 

In our present clinical and molecular diagnosis study, we have identified four families from the Pakhtun ethnic group in Pakistan affected with CMT2EE, CMT4F, CMTX, and ARSACS.

## 2. Method

### 2.1. Samples

All four families were enrolled from the Pakhtun ethnic group of Khyber Pakhtunkhwa province of Pakistan. The pedigrees construction and analysis were conducted in light of standard protocol [33]. Family DG-01 (Figure 1A) segregates two conditions: autosomal recessive CMT and autosomal recessive ARSACS. Family BD-06 (Figure 1B) segregates autosomal recessive CMT. Family MR-01 (Figure 1C) segregates X-linked dominant CMT. Family ICP-RD11 (Figure 1D) segregates autosomal recessive ARSACS. Each study participant provided peripheral blood samples, which were collected and stored in 5 mL EDTA tubes.

### 2.2. Genomic DNA Extraction and Genetic Analysis

Genomic DNA was extracted from all available family members’ blood using PureLinked^®^ genomic extraction kits (Thermo Fisher, Waltham, MA, USA). DNA quantity and quality were analyzed using a Nanodrop 1000 spectrophotometer (Thermo Fisher, Waltham, USA). WES was planned for the index patients in each family for the genes involved in axonal motor sensory neuropathies (Appendix A). The exome, which covers about 22,000 disease-causing human genes, was captured using xGen Exome Research Panel v2 (Integrated DNA Technologies, Coralville, IA, USA) [34,35]. The captured exome was sequenced using NovaSeq 6000 (Illumina, San Diego, CA, USA) with a mean depth of coverage of 212.96. The sequencing data were aligned to the human genome database GRCh37 and the human mitochondrial genome reference database rCRS. The alignment covered approximately 99.3% of the RefSeq protein coding region. An in-house software, EVIDENCE, was used for variant interpretation and prioritization based on patient phenotypes, family history, and clinical diagnosis in accordance with the guidelines of the American College of Medical Genetics and Genomics (ACMG) and the Association for Molecular Pathology (AMP) [34,35,36]. The pathogenicity and segregation of the identified variants were then confirmed by sequencing selected affected and unaffected individuals in the families using forward and reverse primers (Table 2) from Macrogen Inc., Seoul, South Korea (https://dna.macrogen.com/ (accessed on 29 April 2022)).

For primer design, the genomic DNA sequences of the genes were retrieved from the Ensembl genome browser (https://asia.ensembl.org/ (accessed on 2 April 2022)). The following gene sequences were processed:*MPV17* with accession number ENSG00000115204 and transcript ID ENST00000380044.6;*SACS* with accession number ENSG00000151835 and transcript ID ENST00000382292.9;*PRX* with accession number ENSG00000105227 and transcript ID ENST00000324001.8;*GJB1* with accession number ENSG00000169562 and transcript ID ENST00000361726.7.

### 2.3. Protein Modeling of MPV17, GJB1, and SACS

As no known X-ray crystallography-based 3D structures were present, the artificial intelligence-based 3D structures of MPV17^WT^` (AlphaFold ID: AF-P39210-F1) and GJB1^WT^` (AlphaFold ID: AF-P39210-F1) were downloaded from the AlphaFold Protein Structure Database [37,38]. As no known X-ray crystallography-based 3D structure of SACS^WT^ was present, its sequence (uniport ID: Q9NZJ4-1; 831-1740 AA) was subjected to I-TASSER [39,40] for 3D structure prediction. The structures were subjected to homology modeling as a template for predicting the 3D structure of MPV17^GLY28VAL^, GJB1^GLY21ARG^, and SACS^ARG1645PRO^ through MODELLER [41]. The mutated structures were subjected to energy minimization through AMBER ff99SB using UCSF Chimera 1.8.1 [42]. Rotamers and amino acid outliers were removed through WinCoot 0.9.8.1 [43]. The quality of the structure was assessed by MolProbity [44] and VERIFY3D [45]. Finally, the optimized design was visualized and analyzed through USCF Chimera 1.8.1 [42] and PyMOL (http://www.pymol.org/pymol (accessed on 10 January 2023)) for mutational and structural analysis. To assess the pathogenic/deleterious effect of mutations MPV17^GLY28VAL^, GJB1^GLY21ARG^, and SACS^ARG1645PRO^ MTBAN was employed [46].

## 3. Results

### 3.1. Clinical Features of the Patients

#### 3.1.1. Family DG-01

In the first family, DG-01 (Figure 1A), four affected siblings are presented: a 16-year-old boy (V-1) and three girls, a 14-year-old (V-2) and 9-year-old maternal twins (V-3, V-4), born to first cousin normal parents. They had a delayed onset of walking and later developed difficulty walking, gait ataxia, and spasticity of the lower limbs. They developed mild pes cavus at the ages of 16 and 14 years, have abnormal foot phalanges (Figure 2D), and walk sidewise with difficulty lifting their feet, and they frequently fall. They gradually lost their grip and cannot hold anything in their hands or stand without support. Their hands are affected by tremors and developed hand curvature. They are incredibly petite and have developed bilateral scapular dyskinesis and winged scapula (Figure 2A–C). They are intellectually dull compared to their normal siblings and cousins. They have squint strabismus eyes and dry skin that appears as white scales on their legs. The elder siblings presented more severe phenotypes, while the two younger siblings had not developed proper phenotypes. Electromyography (EMG) of V-1 and V-2 of bilateral lower and upper limbs shows spontaneous activities with neuropathic configurations of the myopathic motor unit potential (MUAPs). The electrodiagnostic studies developed for V-1 and V-2 show electrophysiological evidence of demyelinating polyneuropathy. Serum analysis performed for V-1 and V-2 shows severe vitamin D deficiency.

#### 3.1.2. Family BD-06

The second family, BD-06 (Figure 1B), is a large consanguineous family with eight affected individuals (V-1, V-4, V-5, V-6, V-7, V-9, V-12, and V-14) ranging in age from 16 years (V-14) to 36 years (V-1), all born to first cousin marriages. The indexed patient for WES is V-12, a 19-year-old boy. They developed phenotypes at an early age, usually walking at an early age but later being unable to stand and walk properly. From the first decade of life, all affected members of this family developed pes cavus, bilateral steppage gait, and began toe walking. With age, they began to experience extreme difficulties in walking, standing, maintaining balance, areflexia, and sensory ataxia. Their hands, legs, and body are affected by tremors and developed hand curvature, bone deformities of the feet and hands, swollen joints, and abnormal phalanges. They are incredibly petite, have abnormal vertebral phenotypes, and developed bilateral scapular dyskinesis and winged scapula (Figure 2F–K). V-1, V-4, V-5, V-7, V-12, and V-14 were diagnosed with myoclonic seizure and epilepsy. With age, they also showed abnormal eye phenotypes such as ptosis or blepharoptosis, muscular atrophy, and steppage gait. However, mentally, all affected individuals are sound. The severity, signs, and symptoms within the family varied from patient to patient and clade to clade, with even siblings showing different phenotypes. Some members developed progressive sensory hearing loss early, and some are entirely deaf (V-1 and V-4).

#### 3.1.3. Family MR-01

The third family presented here, MR-01 (Figure 1C), has two affected males born to a moderately affected mother (III-4) with no other family history. The index patient (IV-4) is a 28-year-old male, while another sibling, IV-3, is a 32-year-old male. They developed mild CMTX1 phenotypes in the second decade of their lives, becoming more prominent with advancing age. All three affected individuals (III-4, IV-3, IV-4) have developed pes cavus, cleaved toe, upper and lower limb numbness, tremor, weakness, and hyporeflexia (Figure 2L). From early childhood, they were diagnosed with epilepsy, especially when exposed to crowd and pressure. They cannot speak properly, spit while talking, have language delay, irregular and abnormal speech, mental stress, and non-progressive intellectual disability. For the indexed patient (IV-4), EEG was recorded many times, which showed a gross abnormal EEG pattern and suggested tonic–clonic seizure, myoclonic jerk, and epilepsy. Their mother (III-4) developed pes cavus and walking disabilities, upper and lower limb numbness, tremor, weakness, and hyporeflexia. With advancing age, the severity of the phenotypes increased.

#### 3.1.4. Family ICP-RD11

The fourth family presented here (Figure 1D) has two affected siblings. The indexed patient (V-5) is a 26-year-old female born to third cousins with one 35-year-old affected brother. Both patients showed ARSACS phenotypes in the second decade of life, but at the time of blood collection, the affected brother, V-1, was completely paralyzed and unable to stand or walk. They developed pes cavus, astasia, difficulty walking, joint stiffness, joint contractures, and spasticity of the lower limbs. In addition, they have muscular atrophy of the lower limbs, and the upper limbs are comparatively less atrophic (Figure 2E).

### 3.2. Genetic Analysis

WES in family DG-01 identified two homozygous variants: in *MPV17* (NM_002437.5: c.83G>T: NP_002428.1: p.Gly28Val) (Figure 3A–D) and *SACS* (NM_014363.6: c.4934G>C: NP_055178.3: p.Arg1645Pro) (Figure 3I–L right). In family BD-06, WES identified a biallelic variant (c.231C>A NM_181882.3: NP_870998.2: p.Tyr77Ter) (Figure 3E–H) in *PRX*. In family MR-01, WES identified a homozygous variant (NM_000166.6: c.61G>C: NP_000157.1: p.Gly21Arg) (Figure 3M–P) in *GJB1*. In family ICP-RD11, a variant (NM_014363.6: c.262C>T: NP_055178.3: p.Arg88Ter) (Figure 3I–L left) in *SACS* was identified through WES in the index patient. In families DG-01, BD-06, and ICP-RD11, Sanger validation confirmed autosomal recessive segregation of the identified variants with the disease phenotypes, while in family MR-01, Sanger sequencing validated the X-linked dominant mode of segregation. The variants were classified as *MPV17* (c.83G>T) VUS, *SACS* (c.4934G>C) VUS, *PRX* (c.231C>A) pathogenic, *SACS* (c.262C>T) pathogenic, and *GJB1* (c.61G>C) likely pathogenic, according to the ACMG/AMP guidelines. All the variants except ICP-RD11 c.262C>T in SACS were novel and absent from the ClinVar (https://www.ncbi.nlm.nih.gov/clinvar/ (accessed on 5 January 2022), gnomAD v3.1.2 (https://gnomad.broadinstitute.org/ (accessed on 5 January 2022)), and 1000 genomes (https://www.internationalgenome.org/ (accessed on 5 January 2022) databases. Population screening of 200 ethnically matched controls excluded the presence of these variants in healthy individuals.

### 3.3. MPV17 c.83G>T: p.Gly28Val

For most regions, the pLDDT value of the wild-type structure of MPV17 was between >= 90 and >= 70. The 3D-1D score of MPV17^GLY28VAL^ was >= 0.2. MPV17^WT^ contains six α-helices. Due to the induction of mutation in MPV17^GLY28VAL^, α4 extended to one amino acid (ASN112) compared to MPV17^WT^. MTB analysis (score: −8.091; Z-score: 0.2844; probability of deleteriousness: 0.6603) suggested the significant pathogenicity/deleteriousness of the mutation (Figure 4A).

### 3.4. GJB1 c.61G>C: p.Gly21Arg

For most regions, the pLDDT value of the wild-type structure of GJB1 was between >= 90 to >= 70. The 3D-1D score of GJB1^GLY21ARG^ was >= 0.2. GJB1^WT^ contains eight α-helices and two β-sheets. Due to the induction of mutation in GJB1^GLY21ARG^, a slight upward movement of α4 and α5, along with a significant outward movement of the C-terminal (α8), was observed (Figure 4B).

### 3.5. SACS c.4934G>C: p.Arg1645Pro

The structural analysis of the 3D structure of the wild type and mutated SACS revealed that the N-terminal region of SACS^ARG1645PRO^ was significantly different from SACS^WT^. Specifically, inward movement of α1, α2, and α3 was observed, while outward movement of α5, α6, and α7 was observed. Moreover, α11 converted into a loop and a loop between α19 and α20 converted into a helical turn in SACS^ARG1645PRO^ as compared to SACS^WT^ (Figure 4C).

## 4. Discussion

We reported a cohort of 16 patients from four unrelated Pakistani families of Pakhtun descent. All affected individuals showed clinical phenotypes of CMT and ARSACS. Patient DG-01 was diagnosed with a merger of autosomal recessive CMT2EE and ARSACS. Patient BD-06 was diagnosed with autosomal recessive CMT4F, and MR-01 was diagnosed with X-linked dominant CMTX1. While a few cases of CMTEE have been reported globally, this is the first case reported in the Pakistani population. Similarly, there are only a small number of reported cases of ARSACS, CMT4F, and CMTX1 in the literature from Pakistan.

The first diagnostic characteristics of axonal sensorimotor neuropathies are walking disabilities and ataxia [24,26]. Other reported phenotypes in the literature, as compared with our families, are tabulated in Table 1. In our study group, we reported distal upper limb weakness, hand tremor, delayed motor development, ataxia, pes cavus, intellectual disabilities, and speech articulation, with minor phenotypic variations and varying ages of onset, as previously documented [9,13,14,15,16,19,20,21,24,26,30,31,47]. 

In addition to the common phenotypes, in CMT2EE, sensorineural hearing loss, hyporeflexia, areflexia, and scoliosis are notable phenotypes [13,27,28,29] that were present in our family DG-01. In CMT4F, clawed hands [18], distal limb muscle atrophy [19], areflexia [29], and scoliosis [19] have also been reported, which were confirmed in our family BD-06. Additionally, we observed sensorineural hearing loss, a new phenotype not reported in CMT4F. In our family MR-01, we found Achilles tendon contractures, numbness of the hands and legs, hyporeflexia, areflexia, and toe walking, as reported previously [13,16,28,29].

Biallelic variants causing CMT2EE have been found in *MPV17* (OMIM: 137960), a gene located on chromosome 2p23 that produces an inner mitochondrial membrane protein consisting of four transmembrane domains [48,49]. This protein is a non-selective channel responsible for maintaining the balance of nucleotides and mitochondrial DNA within the mitochondria. To date, 56 mutations have been reported in *MPV17*, the majority of which cause a condition called mitochondrial DNA depletion syndrome-6 (MTDPS6: 256810). Only two variants have been reported to cause CMT2EE, including a missense c.122G>A: Arg41Gln, and a splice site c.376-9T>G [21,27,50]. Our study identified a novel biallelic variant in *MPV17* (NM_002437.5), c.83G>T, which changes a highly conserved glycine to valine at amino acid position 28 (NP_002428.1: p: Gly28Val). A missense alteration is a common mechanism for Charcot–Marie–Tooth disease, axonal, type 2EE, and the rate of benign missense variants is relatively low. In silico prediction tools, conservation analysis, and protein simulations predicted that this variant is damaging to the protein structure and function (REVEL: 0.787 >= 0.6, 3CNET: 0.983 >= 0.75) (Appendix A; Figure 4A).

ARSACS (OMIM: 270550) is caused by biallelic modifications in the *SACS* (OMIM: 604490) located on chromosome 13q12.12. This gene encodes for the large multidomain sacsin protein (Q9NZJ4), which acts as a regulator of the Hsp70 chaperone machinery and is responsible for processing ataxin-1 [51]. To date, 328 mutations have been reported in *SACS*, including 180 missense mutations, 3 splice substitutions, 91 small deletions, 37 insertions/duplications, 16 deletions, and 1 complex rearrangement. In our study, we have reported two variants in *SACS*. A recurrent variant c.262C>T (NM_014363.6) in family ICP-RD11, which lies in a highly conserved region of the sacsin protein (NP_055178.3) (Figure 5B left) and creates a premature termination codon (Figure 3I–L left), is expected to cause a loss of functional protein via non-sense-mediated mRNA decay. Additionally, we identified a novel biallelic variant in family DG-01 in *SACS*, c.4934G>C (Figure 3I–L right), at amino acid position 1645, which changes a large hydrophilic arginine to a neutral proline in a highly conserved region of the sacsin protein (Figure 5B right). In silico prediction tools and conservation analysis predicted that this variant is probably damaging to the protein structure/function (REVEL: 0.849 >= 0.6, 3CNET: 0.92 >= 0.75) (Appendix A; Figure 4C). The mutant sacsin protein is non-functional in large neurons, particularly within brain motor systems, including cerebellar Purkinje cells, which are associated with ARSACS.

CMT4F (OMIM: 614895) is caused by biallelic mutations in the *PRX* (OMIM: 605725) located on chromosome 19q13.2. The gene encodes for the Periaxin protein (Q9BXM0), which is necessary for maintaining the myelin sheath, normal nerve impulse, and conduction velocity. *PRX* has seven exons and encodes for two isoforms: L- and S-periaxin protein [31]. To date, a total of 116 mutations have been reported in *PRX*, of which only 10 variations—5 missense, 4 small deletions, and 1 insertion—cause CMT4F. Our analysis of family BD-06 revealed a novel biallelic pathogenic variant in *PRX* (NM_181882.3 c.231C>A) through WES analysis (Figure 3E–H). The c.231C>A variant leads to a premature termination of the peroxin protein at amino acid position 77 (NP_870998.2) (Figure 5C), which further leads to immature protein synthesis or loss of functional protein via nonsense-mediated mRNA decay. Furthermore, in silico prediction tools and conservation analysis predicted that the variant is damaging to the protein structure/function (Appendix A). The lack of periaxin protein in the affected family members can be related to their severe phenotypes.

CMTX1 (OMIM: 302800) is caused by mutations in the *GJB1* (OMIM: 304040) located on the X-chromosome (Xq13.1). This gene encodes for the gap junction beta-1 connexin 32 (CX32) protein [52]. A clear genotype–phenotype correlation in *GJB1* is not fully understood, but it is proposed that the altered protein is degraded quickly and trapped inside the cell, preventing it from reaching the cell membrane and forming gap junctions. The loss of functional gap junctions alters the activity of Schwann cells and the production of myelin. In addition to changes in the peripheral nervous system, *GJB1* mutations are also associated with the loss of myelin in the brain and spinal cord [53,54]. Although the inheritance pattern is X-linked dominant, hemizygous males are more severely affected than heterozygous females [55]. In family MR-01, a hemizygous missense variant c.61G>C in *GJB1* (NM_000166.6) (Figure 3M–P) lies in a highly conserved region of the gap junction beta 1 (P08034) protein (Figure 5D). GJB1 proteins form a closed cluster of six connexin proteins, which are believed to be responsible for the diffusion of low molecular weight materials between neighboring cells. The abnormal EEG pattern can be related to tonic–clonic seizures, myoclonic jerks, and epilepsy. The variant is in a well-established functional domain or exonic hotspot, where pathogenic variants have frequently been reported. A missense variant is a common mechanism associated with Charcot–Marie–Tooth neuropathy and X-linked dominant, 1, and the rate of benign missense variants is relatively low. In silico prediction tools and conservation analysis predicted that this variant was probably damaging to the protein structure/function (REVEL: 0.971 >= 0.6, 3CNET: 0.997 >= 0.75) (Appendix A; Figure 4B).

In conclusion, neuropathies are a diverse group of genetic disorders and have many overlapping phenotypes with many reported genes. Therefore, simple clinical and phenotypic investigation alone makes an exact diagnosis of the neuropathy types impossible. In such cases, WES analysis is the best, most precise, and most time-effective procedure for these diversified genetic neuropathies. In societies such as Pakistan, where around 60% of people practice cousin marriages, the segregation of mutant alleles is much higher [35]. The disease allele burden can be reduced, or at least attempted to be reduced, by premarital testing, newborn genetic disorder screening, and parental genetic screening [56,57]. The disease allele burden can also be reduced by carrier genetic screening, parental counseling, and neonatal work-up if the parents wish for future progeny [35].

## Figures and Tables

**Figure 1 genes-14-00328-f001:**
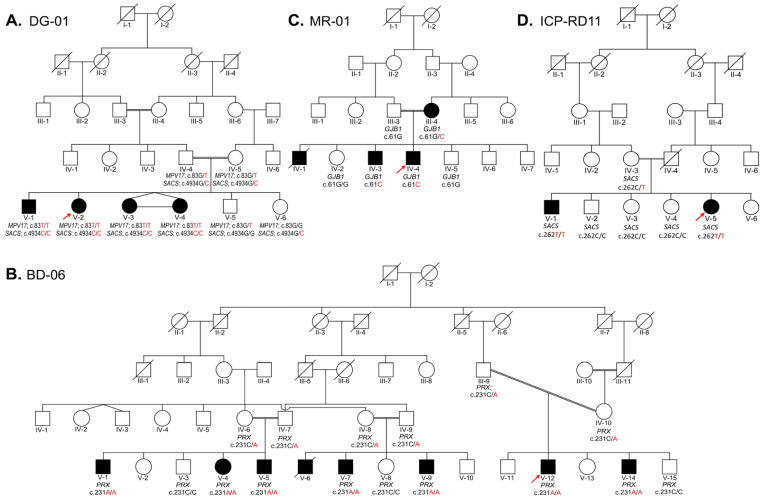
Four families designated as DG-01 (**A**), BD-06 (**B**), MR-01 (**C**), and ICP-RD11 (**D**) are presented here. Family (**A**) segregates two conditions: autosomal recessive CMT type 2EE and autosomal recessive ARSACS. Family (**B**) segregates autosomal recessive CMT Type 4F, while family (**C**) segregates X-linked dominant CMT. Family (**D**) segregates autosomal recessive ARSACS. In each family, the index patient is indicated with an arrow. The genotypes of the family members are written below their symbols, whose samples were available for Sanger sequencing. (**A**) DG-01: In the five-generation pedigree, the four affected siblings (V-1, V-2, V-3, and V-4) are homozygous mutants for both *MPV17* c.83G>T and *SACS* c.4934G>C, while one unaffected sibling (V-5) is homozygous wild type for *MPV17,* and another unaffected sibling (V-6) is homozygous wild type for *SACS*. Their parents (IV-4 and IV-5) are heterozygous carriers for *MPV17* c.83G>T and *SACS* c.4934G>C. (**B**) BD-06: A five-generation pedigree presents the disease in three different clades. Each clade was confirmed for the variant in *PRX*, and Sanger sequencing was carried out and confirmed that affected individuals (V-1, V-4, V-5, V-7, V-9, V-12, and V-14) are homozygous mutants, while parents (IV-6, IV-7, IV-8, IV-9, III-9, and IV-10) are heterozygous carriers, and normal siblings (V-3, V-7, V-15) are homozygous wild type for *PRX* c.231C>A. (**C**) MR-01: In a four-generation pedigree, the two affected siblings (IV-3 and IV-4) are hemizygous mutants, while one unaffected sibling (IV-5) and the father (III-3) are homozygous wild type, and the mother (III-4) is a heterozygous carrier for *GJB1*: c.61G>C variant. (**D**) ICP-RD11: In a five-generation pedigree, the two affected siblings (V-1, V-5) are homozygous for the mutant allele, while three unaffected siblings (V-2, V-3, and V-4) are homozygous for the wild type allele and their mother (IV-3) is heterozygous carrier for the *SACS* c.262C>T variant.

**Figure 2 genes-14-00328-f002:**
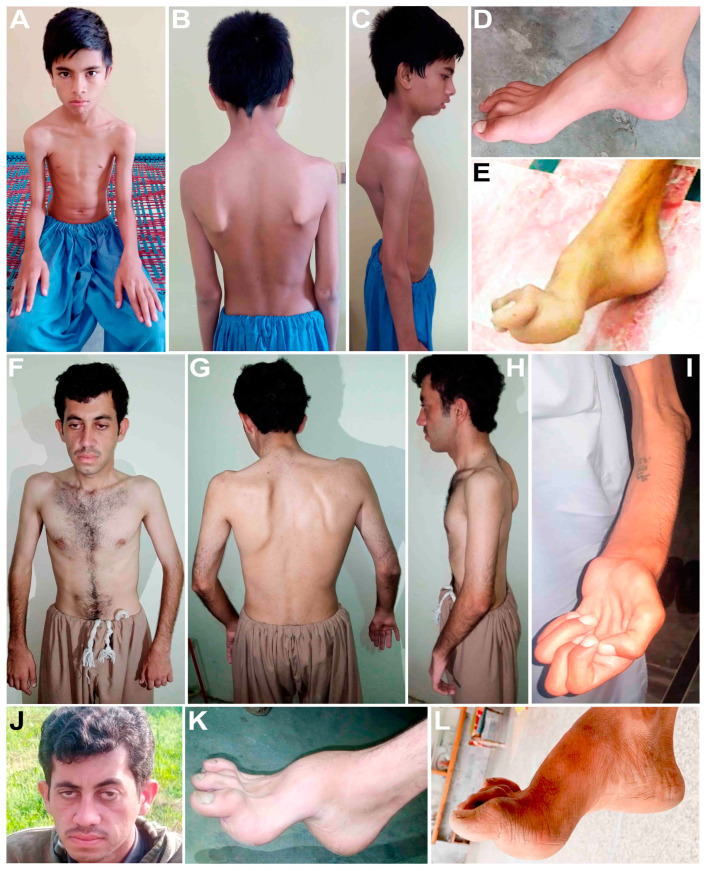
Clinical observations of DG-01 (**A**–**D**), ICP-RD11 (**E**), BD-06 (**F**–**K**), and MR-01 (**L**) are presented here. (**A**) and (**F**) show extreme petite body in family DG-01 and BD-06. (**B**) and (**G**) demonstrate bilateral scapular dyskinesis and winged scapula in family DG-01 and BD-06. (**F**) and (**J**) present abnormal eye phenotypes such as ptosis or blepharoptosis in family DG-01 and BD-06. (**I**) shows clawed hands and muscular atrophy in family BD-06. (**D**,**E**, **K**,**L**) show pes cavus and abnormal phalanges in family DG-01, BD-06, MR-01, and ICP-RD1.

**Figure 3 genes-14-00328-f003:**
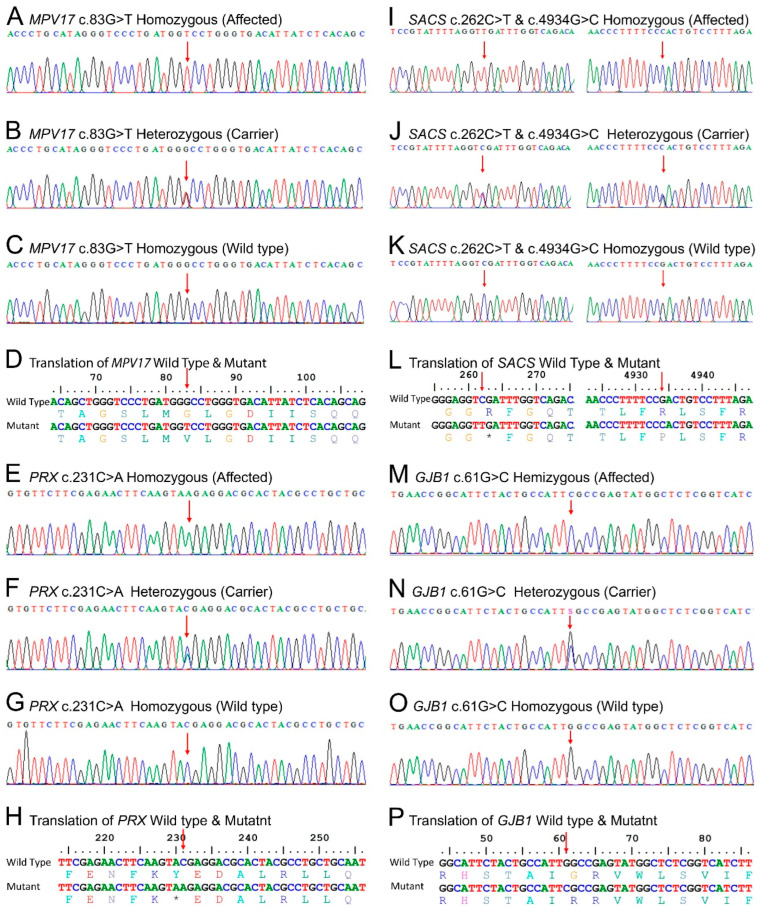
Sanger sequencing trace files were analyzed to identify homozygous mutated, hemizygous mutant, heterozygous carriers, and homozygous wild-type individuals for each gene. The point of alteration is indicated by an arrow above the point of mutation. Figures (**A**–**C**) show the mutation *MPV17* c.83G>T, figures (**E**–**G**) show *PRX* c.231C>A, figures (**I**–**K right**) show *SACS* c.4934G>C, figures (**I**–**K left**) show *SACS* c.262C>T, and figures (**M**–**O**) show *GJB1* c.61G>C. Figures (**D**,**H**,**L**) and (**P**) show the single letter translation of both the wild-type and mutant gene.

**Figure 4 genes-14-00328-f004:**
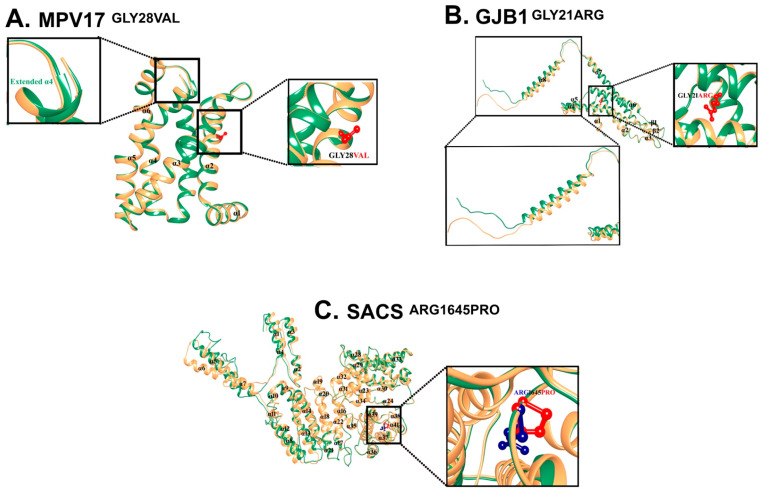
(**A**) Mutational and structural analysis of MPV17. Structural differences due to mutations in MPV17. Superimposition of MPV17^WT^ (sandy brown) and MPV17^GLY28VAL^ (sea green). GLY28 is labeled in black, and VAL28 is shown and labeled in red. Extended-α5 is labeled and shown in sea green. All the α-helices are marked in black. (**B**) Mutational and structural analysis of GJB1. Structural differences due to mutations in GJB1. Superimposition of GJB1^WT^ (sandy brown) and GJB1^GLY21ARG^ (sea green). GLY21 is labeled in black, and ARG21 is shown and labeled in red. The region signifying the structural changes is zoomed in on and displayed in the inset. (**C**) Structural differences due to mutation in SACS. Superimposition of SACS^WT^ (sandy brown) and SACS^ARG1645PRO^ (sea green). ARG shown and labeled in blue, PRO shown and labeled in red. All the α-helices are labeled in black.

**Figure 5 genes-14-00328-f005:**
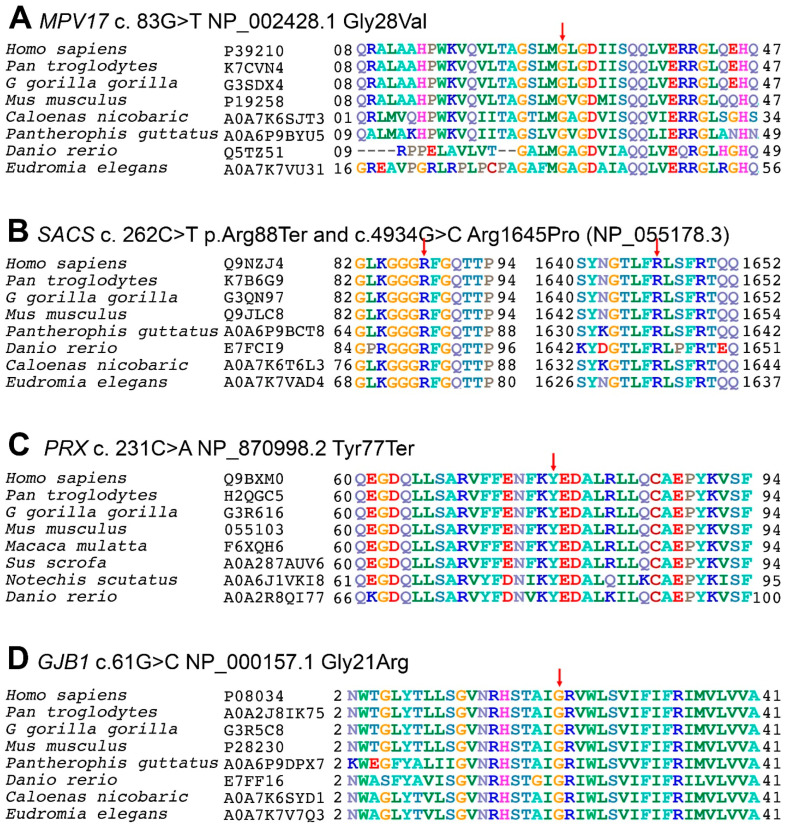
P rotein multiple sequence alignments of inner mitochondrial protein (MPV17: P39210), sacsin (SACS: Q9NZJ4), periaxin (PRX: Q9BXM0), and gap junction beta1 (GJB1: P08034) proteins with their respective members across species. The first column indicates the species and the second is its entry in the UniProtKB database (https://www.uniprot.org/uniprot/ (accessed on 5 January 2022)). The numbers before and after the sequences indicate the respective amino acids in proteins, while the mutated point is shown by an arrow above it. (**A**) In MPV17 (P39210) protein, the glycine at codon 21 is replaced with valine in a highly conserved region of the first transmembrane domain. (**B**) In sacsin (Q9NZJ4), (**B Left**) the arginine shown in blue at codon 88 is replaced with a termination codon in a highly conserved region, which truncates the protein. (**B Right**) The arginine (dark blue) at codon 1645 is replaced by proline in a highly conserved region. (**C**) In PRX (Q9BXM0) protein, the arginine (dark blue) at codon 77 is replaced with a termination codon in a highly conserved region of the first domain. (**D**) In GJB1 (P08034), the glycine (orange) located at codon 21 is replaced with arginine in a highly conserved region of the cytoplasmic topological domain.

**Table 1 genes-14-00328-t001:** Different phenotypes have been reported in literature related to three types of Charcot–Marie–Tooth disease and autosomal recessive spastic ataxia of Charlevoix–Saguenay type. These have been compared with the four gene mutations we have reported.

S/No.	Phenotypes with References Respective to the Disorder	CMT 2EE	CMT 4F	CMT X1	SACS	DG-01	BD-06	MR-01	DG-01	ICP_RD11
1	Sensorineural hearing loss [12,13]	+	-	+	-	+	+	-	+	-
2	Nystagmus [13]	-	-	+	+	-	-	-	-	-
3	Achilles tendon contractures [13]	-	-	+	-	-	-	+	-	-
4	Absent ankle jerk [13,14]	-	-	+	+	-	-	+	+	+
5	Distal upper limb weakness [13,14]	+	-	+	+	+	+	+	+	+
6	Hand tremor [15,16]	-	-	+	+	+	+	+	+	+
7	Clawed hands [17,18]	+	+	-	-	-	+	-	-	-
8	Muscle biopsy showed neurogenic changes [13]	-	-	+	-	n/a	n/a	n/a	n/a	n/a
9	Numbness of hands and legs [16]	-	-	+	-	-	-	+	-	-
10	Delayed motor development [13,19,20]	-	+	+	+	+	+	+	+	+
11	Distal limb muscle atrophy [19,21,22,23]	+	+	+	-	-	+	-	-	+
12	Axonal sensorimotor neuropathies [24,25,26]	+	+	+	+	+	+	+	+	+
13	Hyporeflexia [27,28]	+	-	+	-	+	-	+	+	-
14	Areflexia [29]	+	+	-	-	+	+	-	+	+
15	Ataxia [16,24,30]	+	+	+	+	+	+	+	+	+
16	Walking difficulties [24,25,27]	+	+	+	+	+	+	+	+	+
17	Toe walking [13]	-	-	+	-	-	+	+	-	-
18	Pes cavus [14,21,31]	+	+	+	+	+	+	+	+	+
19	Clawed toe [17]	+	-	-	-	-	+	-	-	-
20	Foot drop [17]	+	-	-	-	-	-	-	-	-
21	Scoliosis [14,19]	+	+	-	+	+	+	-	+	+
22	Intellectual disabilities [26,32]	+	+	+	+	+	-	+	+	+
23	Speech articulation [16,30]	+	+	+	+	+	+	+	+	+
24	Decreased COX reactivity [17]	+	-	-	-	n/a	n/a	n/a	n/a	n/a
25	Reported causative genes	*MPV17*	*PRX*	*GJB1*	*SACS*	*MPV17*	*PRX*	*GJB1*	*SACS*	*SACS*

**Table 2 genes-14-00328-t002:** Primer list with gene name, exon No., annealing temperature, and product size.

S/No	Gene	Exon	Primers (5′-3′)	Temp	Product Size (bp)
1	*MPV17*	Exon 3	F 5′-AAGCCCTGGGGTTCAGAGTA-3′	56 °C	320 bp
			R 5′-TGTGAGAGTCCAAGGGAAGC-3′		
2	*SACS*	Exon 5	F 5′-CACATTGGTGACAATTCATGG-3′	56 °C	373 bp
			R 5′-AGCTTAGCGCATTTCTTTGC-3′		
2	*SACS*	Exon 10	F 5′-CAAATCCAATCCTGGGATCA-3′	57 °C	339 bp
			R 5′-AAACTGGGGTTGGTTTCCTC-3′		
3	*PRX*	Exon 3	F 5′-GGGACTAGCGTAACTGCGACT-3′	55 °C	326 bp
			R 5′-TGATCTCGTAGCCAGACACG-3′		
4	*GJB1*	Exon 1	F 5′-GAGAAGCTGGCAAGGGAGAT-3′	56 °C	773 bp
			R 5′-AGGCAGCTAGCATGAAGACG-3′		

## Data Availability

The raw data supporting the conclusions of this article will be made available by the authors without undue reservation. We have submitted the variants data to ClinVar, and the accession numbers VCV000559870.5, VCV001722539.1, VCV000958213.6, and VCV001722538.1 have been assigned to them.

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
