# Peer review of "Novel Variants in MPV17, PRX, GJB1, and SACS Cause Charcot–Marie–Tooth and Spastic Ataxia of Charlevoix–Saguenay Type Diseases"

_genes, 2023, doi:10.3390/genes14020328_

Round 1
Reviewer 1 Report
In this study, authors identified four families along with 16 affected individuals for clinical and molecular diagnoses. Whole exome sequencing and Sanger sequencing were performed. The WES analysis in an indexed patient identifies two novel variants: c.83G>T (Gly28Val) in MPV17 and c.4934G>C (Arg1645Pro) in SACS. In family ICP-RD11, a recurrent mutation, c.262C>T (Arg88Ter) in SACS, was identified that causes ARSACS. Another novel variant, c.231C>A (Arg77Ter) in PRX, was identified in another, which causes CMT4F. In another family, a hemizygous missense variant c. 61G>C (Gly21Arg), in GJB1 was identified in the indexed patient. This is a well-written and interesting study, but I have few comments before it is accepted for publication.
Comments:
1- Please provide detailed clinical information on the families and affected members in this study. Did the authors find any genotype-phenotype correlations?
2- Please italicize gene names throughout the manuscript.
3- WES protocol has been described very briefly. Kindly elaborate on the procedure of WES.
4- I recommend citing the new release of HGMD, i.e., 2019.4 and mentioning its updated information
5- Mention transcript ID with mutation identified.
6- The detailed Figure legends are not written and need to explain what figure represents what in the revised manuscript.
7- The authors need to improve the figure quality with maximum resolutions.
8- Ethical details missing need a clear statement.
9- Please provide the primer sequence used for the PCR and Sequencing validation in the methodology section.
10- The analysis strategy is not in sequence; it should be in this way fastq files were converted to Bam, then BAM to vcf and then variant filtration was carried out on this file, etc needed in detail.
11- Please add the latest reference related to the study and correct the reference format, as some references are also missing.
12- Language also needs corrections, as some of the sentences in the introduction and discussion sections are unclear.
13- The patient's identity must be hidden by covering the eyes in the figures.
14- The conclusion of the study needs to change with clear output.
Overall, the manuscript is well written; however, it needs improvement before the final acceptance for publication.
Author Response
Response to the reviewer’s comments 1
In this study, authors identified four families along with 16 affected individuals for clinical and molecular diagnoses. Whole exome sequencing and Sanger sequencing were performed. The WES analysis in an indexed patient identifies two novel variants: c.83G>T (Gly28Val) in MPV17 and c.4934G>C (Arg1645Pro) in SACS. In family ICP-RD11, a recurrent mutation, c.262C>T (Arg88Ter) in SACS, was identified that causes ARSACS. Another novel variant, c.231C>A (Arg77Ter) in PRX, was identified in another, which causes CMT4F. In another family, a hemizygous missense variant c. 61G>C (Gly21Arg), in GJB1 was identified in the indexed patient.
This is a well-written and interesting study, but I have few comments before it is accepted for publication.
Comments:
1- Please provide detailed clinical information on the families and affected members in this study. Did the authors find any genotype-phenotype correlations?
Response: Yes, the phenotypes noted in the affected individuals are clear manifestations of the identified variants in the genes.
2- Please italicize gene names throughout the manuscript.
Response: All the genes were italicized throughout the manuscript and preceding word “gene” were removed.
3- WES protocol has been described very briefly. Kindly elaborate on the procedure of WES.
Response: Detailed process of WES has been added
4- I recommend citing the new release of HGMD, i.e., 2019.4 and mentioning its updated information
Response: Note: I don’t have the access to the professional HGMD
5- Mention transcript ID with mutation identified.
Response: Mentioned below the primer table.
6- The detailed Figure legends are not written and need to explain what figure represents what in the revised manuscript.
Response: Legends have been updated with respect to figure details
7- The authors need to improve the figure quality with maximum resolutions.
Response: The figures are revised in high resolutions
8- Ethical details missing need a clear statement.
Response: Missing ethical details has been added in the revised manuscript.
9- Please provide the primer sequence used for the PCR and Sequencing validation in the methodology section.
Response: A table 2, has been added. Thanks for corrections.
10- The analysis strategy is not in sequence; it should be in this way fastq files were converted to Bam, then BAM to vcf and then variant filtration was carried out on this file, etc needed in detail.
Response: Some detail of WES has been added and protein modelling process has been reallocated to methodology section.
11- Please add the latest reference related to the study and correct the reference format, as some references are also missing.
Response: References has been updated, and formatted according to MDPI format.
12- Language also needs corrections, as some of the sentences in the introduction and discussion sections are unclear.
Response: The paper has been thoroughly checked by English professor, and necessary changes has been made. Thanks for suggestions.
13- The patient's identity must be hidden by covering the eyes in the figures.
Response: In case to explain certain phenotypes like “ptosis” pictures of the eyes are important, and in this regard, prior written permission has been taken from the families for publishing their pictures.
14- The conclusion of the study needs to change with clear output.
Response: A conclusive paragraph has been added as suggested.
Overall, the manuscript is well written; however, it needs improvement before the final acceptance for publication.
Response: Thanks for the reviewer’s positive comments.

Reviewer 2 Report
The authors identified causative mutations of MPV17, PRX, GJB1 and SACS in the Pakistani families with CMT and ARSACS. Even though no in vitro or in vivo experiments were performed, their pathogenicity was fairly confirmed. However, it seems that novelty was overally not high and several points are needed to be corrected.
So many typos and redundant expressions were found through the text. Particularly, when the authors use a full word with abbreviation, then use the abbreviation alone. For the legends of figure 1 and 4, similar sentences were repeated many times. Use same expressions to the figure titles before explanation of (A), (B) … Followings are just several examples to be corrected:
[Page 1, Line 32; Page 33] For “Spastic Ataxia Charlevoix-Saguenay type”, it is recommended to edit as followed: autosomal recessive spastic ataxia Charlevoix-Saguenay type (ARSACS).
[Page 2, Line 54-55] Charcot-Marie-Tooth disease (CMT) and Spastic Ataxia Charlevoix-Saguenay type. > CMT and ARSACS.
[Page 2, Line 56] For the keywords, MPV17, PRX, GJB1, and SACS > italic
[Page 2, Line 64] For “in the peripheral nervous system central nervous system”, is it correct expression?
[Page 4, line 104] and another is V-6 is homozygous… ?
This study identified causative mutations of MPV17, PRX, GJB1 and SACS in four Pakistani consanguineous families by WES. Generally, WES exhibits many homozygous variants with uncertain significance in the consanguineous family members. What kinds of rare variants were observed in each families? I would like to recommend to provide a supplementary table including these VUS.
For the references, citations within text and reference list were not prepared according to the journal guideline. The guideline requires a numerical array, for an example, “(Tooth 1886, Charcot, and Marie 1886)” must be corrected to [1,2].
For figure 1, genotypes of family members are provided below their symbols, the letter size is too small to read. Please size-up them. Or, rather, it is recommended to provide a table which describes all the pathogenic mutations.
The study was approved by the bioethical committee of Islamia College Peshawar, according to Helsinki’s declaration 2013. Please provide approval numbers together. In addition, Figure 2 includes several face pictures of affected individuals. Did they provide written consent for this condition? You may also state consent in the text.
It seems that manuscript editing is necessary by a native biological researcher.
Author Response
Response to the reviewer’s comments 2
The authors identified causative mutations of MPV17, PRX, GJB1 and SACS in the Pakistani families with CMT and ARSACS. Even though no in vitro or in vivo experiments were performed, their pathogenicity was fairly confirmed. However, it seems that novelty was overally not high and several points are needed to be corrected.
So many typos and redundant expressions were found through the text. Particularly, when the authors use a full word with abbreviation, then use the abbreviation alone. For the legends of figure 1 and 4, similar sentences were repeated many times. Use same expressions to the figure titles before explanation of (A), (B) … Followings are just several examples to be corrected:
Response: The legend has been updated considering the reviewer comment, and the redundancy has been reduced where possible. The abbreviation is used instead of full form.
[Page 1, Line 32; Page 33] For “Spastic Ataxia Charlevoix-Saguenay type”, it is recommended to edit as followed: autosomal recessive spastic ataxia Charlevoix-Saguenay type (ARSACS).
Response: Accepted and rewritten. Thanks for the corrections.
[Page 2, Line 54-55] Charcot-Marie-Tooth disease (CMT) and Spastic Ataxia Charlevoix-Saguenay type. > CMT and ARSACS.
Response: Accepted and rewritten. Thanks for the corrections.
[Page 2, Line 56] For the keywords, MPV17, PRX, GJB1, and SACS > italic
Response: Accepted and rewritten. Thanks for the corrections.
[Page 2, Line 64] For “in the peripheral nervous system central nervous system”, is it correct expression?
Response: Rectified. Thanks for the corrections.
[Page 4, line 104] and another is V-6 is homozygous… ?
Response: Rectified. Thanks for the corrections.
This study identified causative mutations of MPV17, PRX, GJB1 and SACS in four Pakistani consanguineous families by WES. Generally, WES exhibits many homozygous variants with uncertain significance in the consanguineous family members. What kinds of rare variants were observed in each families? I would like to recommend to provide a supplementary table including these VUS.
For the references, citations within text and reference list were not prepared according to the journal guideline. The guideline requires a numerical array, for an example, “(Tooth 1886, Charcot, and Marie 1886)” must be corrected to [1,2].
Response: References has been updated according to MDPI format. Thanks
For figure 1, genotypes of family members are provided below their symbols, the letter size is too small to read. Please size-up them. Or, rather, it is recommended to provide a table which describes all the pathogenic mutations.
Response: The figure 1 has been updated, the text are made larger and clear as suggested by the reviewer. Thanks for the suggestions.
The study was approved by the bioethical committee of Islamia College Peshawar, according to Helsinki’s declaration 2013. Please provide approval numbers together. In addition, Figure 2 includes several face pictures of affected individuals. Did they provide written consent for this condition? You may also state consent in the text.
Response: In case to explain certain phenotypes like “ptosis” pictures of the eyes are important, and in this regard, prior written permission has been taken from the families for publishing their pictures. Thanks for the corrections.
It seems that manuscript editing is necessary by a native biological researcher.
Response: The revised manuscript has been thoroughly checked by English professor, and necessary changes has been made. Thanks for suggestions.

Round 2
Reviewer 1 Report
The authors inculcated the required changes and I will accept the manuscript for publication in its current form.
Author Response
We are thankful for the reviewer to accept the changes in the revised manuscript. Thanks for the corrections to improve the quality of the study.
Reviewer 2 Report
The authors addressed properly to the most comments, But, no reply was provided for the following query:This study identified causative mutations of MPV17, PRX, GJB1 and SACS in four Pakistani consanguineous families by WES. Generally, WES exhibits many homozygous variants with uncertain significance in the consanguineous family members. What kinds of rare variants were observed in each families? I would like to recommend to provide a supplementary table including these VUS.
Minor comments
[Abstract] For Gly28Val, Arg1645Pro, and Arg88Ter, insert "p." to each variant (ex: Gly28Val p.Gly28Val).
[Page 2, line 75] Charcot-Marie-Tooth disease (CMT) > CMT.
[page 65, line 104-108] It is recommended to delete "(A) DG-01: In the five-generation pedigree, ...... carrier for the SACS c.262C>T variant." These are redundant with results part.
[page 10, line 274] Whole exome sequencing (WES) > WES.
[Page 12, line 337-338] Charcot -Marie-Tooth disease (CMT) and Autosomal Recessive Spastic Ataxia, Charlevoix -Saguenay type (ARSACS) > CMT and ARSACS.
Author Response
The authors addressed properly to the most comments, But, no reply was provided for the following query: This study identified causative mutations of MPV17, PRX, GJB1 and SACS in four Pakistani consanguineous families by WES. Generally, WES exhibits many homozygous variants with uncertain significance in the consanguineous family members. What kinds of rare variants were observed in each families? I would like to recommend to provide a supplementary table including these VUS.
Response: Criteria for variants prioritization (a) autosomal recessive model (b) homozygosity or compound heterozygosity (c) phenotypic relevancy as mentioned in Human Phenotype Ontology and MedGen databases (d) pathogic effect as predicted by SIFT, Polyphen 2 and MutationTaster (e) minor allele frequency matching the rare disease definitions i.e. 1/1000 at least or 0.001 MAF in world population.
This criteria along with the comparison to known variants already mentioned in ClinVar database and those identified by EVIDENCE software of 3Billion.Inc South Korea, retrieved only the single variants mentioned in each family as written in the article. These variants were further validated via Sanger Sequencing in each family. Exclusion of 200 healthy controls ensured that these variants were pathogenic and were only present in the affected families and not in the healthy population.
Minor comments
[Abstract] For Gly28Val, Arg1645Pro, and Arg88Ter, insert "p." to each variant (ex: Gly28Val p.Gly28Val).
Response: Corrected as kindly suggested.
[Page 2, line 75] Charcot-Marie-Tooth disease (CMT) > CMT.
Response: Corrected as suggested.
[page 65, line 104-108] It is recommended to delete "(A) DG-01: In the five-generation pedigree, ...... carrier for the SACS c.262C>T variant." These are redundant with results part.
Response: Deleted
[page 10, line 274] Whole exome sequencing (WES) > WES.
Response: Done corrections as suggested.
[Page 12, line 337-338] Charcot -Marie-Tooth disease (CMT) and Autosomal Recessive Spastic Ataxia, Charlevoix -Saguenay type (ARSACS) > CMT and ARSACS.
Response: Done, thanks for the corrections to improve the manuscript.
